# Chimeric Antigen Receptor T-Cell Therapy for Glioblastoma

**DOI:** 10.3390/cancers15235652

**Published:** 2023-11-30

**Authors:** Kun Ma, Ping Hu

**Affiliations:** 1Queen Mary School, Jiangxi Medical College, Nanchang University, Nanchang 330001, Jiangxi, China; mk939733121@163.com; 2Institute of Translational Medicine, Nanchang University, Nanchang 330001, Jiangxi, China

**Keywords:** glioblastoma, CAR T-cell therapy, clinical trial, tumor heterogeneity, tumor microenvironment, oncolytic virus

## Abstract

**Simple Summary:**

Glioblastoma (GBM) is a highly aggressive brain tumor with low survival rates and limited treatment options. This review explores the potential of chimeric antigen receptor (CAR) T-cell immunotherapy as a promising approach for treating GBM. While CAR T-cell therapy has shown success in blood malignancies, it faces challenges when applied to brain tumors, such as tumor heterogeneity and the inhibitory tumor microenvironment. The review discusses existing research and future prospects for CAR T-cell immunotherapy in glioblastoma, highlighting the need for innovative strategies to improve outcomes for patients. Overall, this review provides insights into the current state and future directions of immunotherapy for GBM.

**Abstract:**

Glioblastoma (GBM), the most common primary brain tumor in adults, is characterized by low survival rates and a grim prognosis. Current treatment modalities, including extensive surgical resection, chemotherapy, and radiation therapy, often yield limited success due to the brain’s sensitivity, leading to significant side effects. Exciting advancements in immunotherapy have recently shown promise in treating various types of tumors, raising hopes for improved outcomes in brain tumor patients. One promising immunotherapy approach is chimeric antigen receptor (CAR) T-cell therapy, which recognizes surface proteins on targeted tumor cells and redirects cytotoxicity towards specific targets. This review aims to discuss the existing research and future prospects for CAR T-cell immunotherapy in treating glioblastoma.

## 1. Introduction

### 1.1. Current Treatments in GBM

Cancer mortality rates have shown a decline over the years across various cancer types, primarily due to advancements in screening and treatment options [1]. However, this progress contrasts with glioblastoma multiforme (GBM), where the five-year survival rate has only marginally increased from 4% to 7% during the same period. This limited improvement can be attributed to the challenges in early detection and treatment. GBM stands as the most prevalent type of primary brain tumor in adults, with an annual incidence of 3–5 per 100,000 people and an overall survival rate of less than 30% [2,3]. In children and adolescents, brain and other nervous system tumors contribute significantly to cancer-related deaths with a generally poor prognosis [1]. The average survival period after diagnosis is less than six months [4].

Current treatment approaches for brain tumors predominantly involve maximal surgical resection, chemotherapy, and radiation therapy [5,6]. However, due to the sensitivity of the brain, these methods seldom result in a cure and often entail significant side effects. Even with standard treatment protocols, the median survival period following the diagnosis is only 14 to 16 months [3]. Acknowledging the constraints of existing treatments underscores the urgent requirement to explore and develop innovative strategies for managing brain tumors.

In recent years, immunotherapy, specifically CAR T-cell immunotherapy, has exhibited significant progress in treating hematologic malignancies. Initially deployed against CD19-positive blood malignancies like lymphoma and acute lymphoblastic leukemia (ALL), CAR T-cell therapies targeting CD19 have gained clinical approval from the U.S. Food and Drug Administration [7]. Notably, Novartis’ Kymriah has demonstrated efficacy in B-ALL, achieving an overall response rate of 81% within three months [8]. Despite these successes, CAR T-cell therapy faces hurdles in treating brain tumors, marked by challenges such as tumor heterogeneity and the inhibitory tumor microenvironment. Here, we explore the existing research and future prospects for CAR T-cell immunotherapy in treating glioblastoma.

### 1.2. History and Structural Evolution of CAR

Chimeric antigen receptors (CARs) are genetically engineered immune receptors designed to identify specific surface proteins on tumor cells. This recognition empowers CAR T-cell therapy to redirect T-cell cytotoxicity towards these defined targets [9]. The CAR is composed of three main parts: an intracellular domain, a transmembrane domain, and an extracellular domain [10,11]. The first generation of CARs includes the CD3zeta chain in intracellular domain for T-cell signal transduction, while second and third generations include one or two co-stimulatory molecules typically derived from CD28 or 4-1BB [12,13,14]. These co-stimulatory molecules provide strong anticancer cytotoxicity to CAR T cells, increase cytokine production, and enhance their proliferation and resistance [15,16]. Evolving from this, fourth-generation CARs integrate additional biomolecules, such as cytokines, to further enhance the antitumor efficacy of T cells [17]. With continuous advancements in CAR technology, these synthetic immune receptors hold significant promise for the development of more effective cancer therapies.

## 2. Preclinical and Clinical Researches of CAR T Cells

### 2.1. Preclinical Trials in Murine Models

CAR T-cell therapy has demonstrated compelling antitumor effects in a mouse model of GBM [18,19,20]. The mouse GBM model is commonly established by subcutaneous or intracranial injection of human tumor cell lines or certain types of primary human cells into immunodeficient mice lacking a fully functional immune system [21]. This allows for sustained engraftment of human cells and provides a model for the development of CAR T-cell therapy targeting GBM. In a preclinical study targeting CD317, CD317-specific CAR T cells exhibited strong anti-GBM activity by specifically targeting several GBM cell lines in vitro, as well as primary patient-derived cells with varying levels of CD317 expression. Furthermore, CD317-specific CAR T cells controlled tumor growth and prolonged survival in an in situ mouse model of GBM, leading to partial cure in some animals treated with CAR T cells [18]. Another preclinical study demonstrated that CAR T cells reduced the number of glioma stem cells and significantly suppressed tumor cell growth in a xenograft tumor model. After systemic administration of these cells, there were no apparent off-target effects or T-cell infiltration in major organs [19].

### 2.2. Completed and Ongoing Clinical Trials

Completed clinical trials of CAR T-cell therapy for glioblastoma have primarily focused on adult and pediatric patients with recurrent or progressive tumors. These trials have successfully identified several potential target antigens, including IL13Rα2, EGFRvIII, and GD2. The selection criteria for these antigens prioritize their low expression in normal brain tissue while exhibiting high expression in GBM. This approach is crucial to minimize the risk of on-target toxicity, which occurs when the targeted antigen is also present in healthy tissue. For instance, in a clinical trial, two patients experienced grade 2 cytokine release syndrome and pulmonary edema due to CAR T-cell infusion targeting EphA2, which is expressed in normal lung tissue epithelial cells, highlighting the importance of careful antigen selection to mitigate such adverse effects [22].

To ensure the safety and efficacy of CAR T-cell therapy, various factors must be taken into account. One such factor is the optimal cell dosage, which can influence the persistence and activity of CAR T cells. Additionally, identifying tumor-specific antigens is essential to further enhance the specificity of treatment. Completed clinical trials have played a pivotal role in providing valuable insights into the safety and bioactivity of CAR T-cell therapy in glioblastoma. These insights will inform the development and refinement of future therapeutic strategies in treating this challenging disease. Table 1 provides a comprehensive summary of several completed phase I and II clinical trials that have utilized CAR T cells for the treatment of human glioblastomas. Furthermore, ongoing trials highlighted in Table 2 demonstrate the continuous efforts to expand treatment options and improve patient prognosis.

#### 2.2.1. IL13Rα2-Targeted CAR T Cells

IL-13 regulates inflammation and immune responses by binding to IL13Rα1. IL-13 also exhibits high affinity binding to the receptor IL13Rα2 [29]. IL13Rα2 exhibits high expression specifically in glioblastoma cells, with absence in normal brain tissue and other healthy tissues [30,31]. This unique expression profile, along with its high-affinity binding to IL-13, renders IL13Rα2 an appealing target for CAR T-cell therapy. The initial human trial (NCT00730613) of IL13Rα2-redirected CAR T cells in 2015 primarily focused on assessing the therapy’s safety and feasibility. Three patients with recurrent glioblastomas received intracavitary or intratumoral infusions. Among them, two demonstrated a transient anti-glioblastoma response, marked by a notable reduction in IL13Rα2 expression in tumor cells, increased tumor necrosis volume observed on MRI scans, and the presence of T cells in tumor microfoci. Despite these promising initial outcomes, all patients experienced tumor recurrence and succumbed to the disease. Tissue analysis from one patient revealed reduced IL13Rα2 expression after CAR T-cell treatment. Intriguingly, tumor cells with low IL13Rα2 expression evaded treatment with IL13Rα2-redirected CAR T cells and continued to proliferate, contributing to tumor recurrence [23].

An ongoing clinical trial (NCT02208362) focusing on CAR T-cell therapy targeting IL13Rα2 for glioblastoma has demonstrated regression of all intracranial and distant spinal tumors after treatment, with this response lasting for 7.5 months. Despite the initial encouraging outcome, disease progression eventually occurred, notably during infusion cycles 7 to 11. Significant increases in cytokine levels, including IFNγ, TNFα, IL-2, and IL-10, was observed in patients’ cerebrospinal fluid after each CAR T-cell infusion. These findings underscore the critical need for closely monitoring cytokine levels during CAR T-cell therapy to mitigate the risk of adverse effects associated with excessive cytokine release [24].

#### 2.2.2. EGFRvIII-Targeted CAR T Cells

Epidermal growth factor receptor (EGFR), a tyrosine kinase receptor, undergoes gene amplification in about 40% of newly diagnosed GBM patients [32]. Additionally, roughly half of EGFR-amplified GBM patients harbor EGFRvIII, the most prevalent genetic mutation in glioblastoma [33,34]. Due to its restricted tumor expression and carcinogenic properties, EGFRvIII has emerged as one of the most attractive CAR targets in GBM research.

The first human trial employing EGFRvIII-redirected CAR T cells (NCT02209376) was conducted in 2017, involving 10 patients received a single intravenous infusion of CAR T cells. After a single infusion, two participants displayed radiographic evidence of disease progression, potentially linked to decrease antigen expression in five patients, accompanied by increased inhibitory molecular and regulatory T-cell infiltration. Notably, the administration of CAR T-modified T cells proved safe and feasible, with no observed tumor-related toxicity or cytokine release syndrome [25]. In contrast to the first human trial, another clinical trial (NCT01454596) investigating anti-EGFRvIII-mediated CAR T-cell therapy resulted in severe adverse effects. Following the CAR T-cell infusion, two patients experienced severe hypoxia, and one fatality occurred after receiving the highest dose of cells. Consequently, this phase I pilot trial assessing anti-EGFRvIII T cell administration to GBM patients failed to demonstrate clinical significance [26]. In addition, preclinical studies have shown that EGFRvIII-specific CAR T-cell therapies demonstrated potent antitumor activity in mouse models [35,36,37]. These CAR T cells are able to lyse GL261/EGFRvIII cells in a dose-dependent manner, promote CD8+ T-cell infiltration, and effectively inhibit the growth of heterogeneous GBM tumors [35]. In another preclinical trial, EGFRvIII CAR T cells were found to significantly prolong the survival of tumor-bearing mice [36]. Rechallenge experiments further revealed that the growth of both EGFRvIII-positive cells and their parental cells was inhibited in cured mice, indicating that EGFRvIII CAR T cells generated antitumor memory responses [35,36].

#### 2.2.3. HER2 Targeted CAR T Cells

Human epidermal growth factor receptor 2 (HER2) is a tyrosine kinase receptor expressed in approximately 80% of glioblastomas [38,39]. However, HER2 is expressed at low levels in various normal tissues, including the lungs [40]. In 2017, the first human trial (NCT01109095) of HER2-specific CAR modified virus-specific T cells was conducted to assess the therapy’s safety and T-cell persistence. Seventeen patients with progressive glioblastoma received one to six intravenous infusions of CAR T cells. Following the infusion, there were notable outcomes observed. Moreover, the infusion of cells was well tolerated by all patients, as no dose-limiting toxic effects were reported. Particularly noteworthy was the prolonged persistence of HER2-CAR VSTs, with detectable CAR T cells still present in the peripheral blood even after 12 months of transfusion [27]. However, the development of HER2-specific CAR T cells is hindered by the low expression of HER2 in normal tissues. Unfortunately, one patient died after receiving a high-dose infusion of HER2-CAR T cells, possibly due to the immediate localization of a large number of administered cells in the lungs. This localization triggered cytokine release through recognition of the low-level HER2 on lung epithelial cells, leading to respiratory distress and pulmonary edema [41].

#### 2.2.4. GD2-Targeted CAR T Cells

GD2 disialogangliosides, primarily found on the cell membranes of normal neurons, are highly expressed in various malignancies, including diffuse pontine glioma (DIPG) [42]. In 2017, the first human trial (NCT03170141) evaluating GD2-specific CAR-mediated T cells aimed to assess the safety and feasibility of this therapy. After the infusion, a significant decrease in GD2 expression was observed. In situ analysis revealed substantial infiltration of T cells and macrophages, with a higher proportion of infiltrated CD8+ T cells compared to pre-infusion tumor specimens. Furthermore, the infusion of GD2-specific 4SCAR T cells reshaped the inhibitory immune microenvironment, leading to reduced infiltration of macrophages in post-infusion tumor specimens. Elevated levels of IL-6 and TNFα were detected in the peripheral blood of all patients post-infusion, with 50% showing at least a 10-fold increase in these cytokines. Additionally, circulating levels of inflammatory cytokines in cerebrospinal fluid significantly surpassed baseline levels prior to infusion [28].

#### 2.2.5. EphA2 Targeted CAR T Cells

EphA2, known for its overexpression in glioblastoma, plays a role in enhanced tumorigenesis and metastasis [43]. The first human trial involving EphA2-redirected CAR T cells (NCT03423992) in 2021 indicated disease progression in two participants, while one showed no signs of deterioration post-infusion. Remarkably, the cells can still be detected in peripheral blood four weeks after transfusion, akin to HER2-specific CAR T cells. Two patients experienced grade 2 cytokine release syndrome with pulmonary edema following CAR T-cell infusion. However, no other organ toxicities, including neurotoxicity, were observed [22]. In conclusion, preliminary findings suggest that this therapy is generally well tolerated and demonstrates short-term clinical efficacy.

The clinical trials mentioned above primarily used two methods for delivering CAR T cells: systemic delivery and local delivery. Unlike hematologic malignancies, brain tumors are regional and rarely metastasize outside the central nervous system. Additionally, the blood–brain barrier (BBB) impedes CAR T-cell infiltration into glioblastoma [25,27]. To improve trafficking and tumor infiltration, local delivery methods like intratumoral/intracavitary (ICT) and intraventricular (ICV) administration are favored. These approaches allow for circumventing the BBB and enable direct delivery of CAR T cells to the tumor site. Local delivery has demonstrated superior antitumor activity and long-term survival benefits compared to intravenous (IV) delivery [44,45]. In situ GBM models treated with IL13Rα2-CAR T cells via ICT showed long-term survival, while IV delivery did not provide significant benefit compared to mock-transduced T cells [45]. Furthermore, local delivery limits off-target toxicity to other systemic tissues during the first pass through the lungs, whereas systemic delivery limits T-cell infiltration within the tumor and may cause lethal pulmonary toxicity [26,46]. Additional clinical testing is necessary to ascertain the impact of different delivery routes on safety and patient outcomes.

The summarized clinical studies indicate that CAR T cells for glioblastoma maintain an acceptable safety profile. However, there are still challenges associated with this therapy, including the low persistence of CAR T cells [26,27], short duration of clinical remission [23,24,25,26,27], limited migration of CAR T cells to the brain [25], tumor heterogeneity [23,24,25], on-target, off-tumor effects [41], and an inhibitory immune microenvironment [25]. The therapeutic impact on glioblastoma falls below expectations compared to their efficacy in treating hematologic malignancies. This disparity may be attributed to both the abundance of CAR T cells and the unique properties of glioblastoma that contribute to its resistance.

## 3. Prospects of CAR T Cells

### 3.1. CAR T-Cell Design

#### 3.1.1. Targeting Multiple Antigens

To mitigate antigen escape arising from intratumoral heterogeneity, the conventional single-targeted CAR design is expanded to incorporate two or more antigens (Figure 1a). This modification ensures that T-cell activation is triggered upon recognition of any one or more antigens. Such an approach substantially enhances tumor cell kill coverage, thereby preventing or delaying antigen escape.

A bispecific CAR (BiCAR) is a bivalent CAR T cell that co-expresses two CARs targeting different antigens on tumor cells. A groundbreaking approach was developed employing the DAP10 interaction with native NKG2D to specifically target cancer cells expressing the NKG2D ligand. This system incorporates GPC3 single-stranded variable fragments (scFv) to create a dual-antigen targeting system. The GPC3-DAP10 CAR demonstrated remarkable efficacy in eliminating heterogeneous cancer cells in vitro and in vivo [47]. This innovative dual-targeting system enhances T-cell killing capacity against cancer cells and broadens the spectrum of tumors that can be recognized, presenting a promising strategy for the clinical treatment of glioblastoma.

Tandem CAR (TanCAR) is another type of bivalent CAR T cell used in cancer biology research. It features two antigen-binding domains connected as tandem extra-CAR domains, sharing a common intracellular signal transduction domain. This unique configuration enables the TanCAR T cells to be activated by encountering either one or two different antigens. In mouse models of GBM, TanCAR T cells targeting HER2 and IL13Ra2 have demonstrated the ability to mitigate antigen escape, resulting in enhanced antitumor efficacy and improved survival outcomes [48]. Moreover, these cells can simultaneously target EGFRvIII and IL-13Rα2, two well-characterized tumor antigens highly expressed on the surface of GBM cells. In vitro experiments illustrate that TanCAR T cells exhibit superior cytotoxicity against heterogeneous glioblastomas [49]. A novel TanCAR T cell designed to target both IL13 Ra2 and EphA2 can effectively recognize and eliminate glioblastoma tumor cells when encountering IL13Ra2 or EphA2 alone, as well as both targets simultaneously. Crucially, this targeted killing does not affect normal cells carrying only IL13Ra1/IL4Ra receptors, achieving tumor regression [50]. Compared to CAR T cells targeting a single antigen or corresponding bispecific CAR T cells, TanCAR T cells demonstrate superior efficacy and selectivity in killing gliomas. They hold significant potential in preventing antigen escape and reducing off-target cell toxicity, offering promising prospects for the treatment of gliomas treatment [48,49,50].

Building upon these advancements, trivalent CAR T cells have been developed to simultaneously target HER2, IL13Rα2, and EphA2. This approach aims to surmount inter-patient variability by addressing multiple antigens. Trivalent CAR T cells, in comparison to monospecific and bispecific CAR T cells, demonstrate enhanced cytotoxicity and cytokine release. Moreover, they foster the development of powerful immune synapses and polarized microtubule tissue centers during interactions with tumor targets. In preclinical studies using autologous GBM patient-derived xenografts (PDX), low-dose trivalent CAR T cells have demonstrated control over established tumors and improved survival in treated animals. However, in a subset of patients, approximately 20% exhibit IL-13Ra2-HER2-EphA2 triple-negative cells, leading to antigen escape. Addressing this challenge through further research is imperative to optimize the efficacy of TanCAR T-cell therapy [51].

Utilizing a bispecific T-cell engager (BiTE) in monovalent or multivalent CAR T cells represents another innovative strategy to address tumor antigen heterogeneity and antigen loss. BiTE, a bispecific antibody, redirects T cells to tumor cells expressing specific antigens by linking two single-stranded variable fragments (scFv): one that recognizes tumor antigens and the other that recognizes CD3 molecules on T cells [52,53]. Thus, BiTE enables T cells to induce cytotoxicity not only towards antigen-expressing tumor cells but also through bystander T cells [54]. When incorporated into CAR T cells, BiTE enhances their antitumor activity against heterogeneous tumors. For instance, EGFR-targeted BiTE-producing CAR T cells exhibit potent and specific antitumor effects, with local BiTE secretion avoiding on-target toxicity [55]. Another variation, BiTE-secreting CAR T cells, effectively targets and eliminates solid tumors expressing different levels of HLA-G and/or PD-L1, particularly those with high simultaneous expression of both targets. This strategy enables specific targeting of solid tumors while recruiting bystander effector cells to attack antigen-deficient cancer cells through Nb-BiTE [54]. In comparison to traditional CAR T cells, BiTE T cells demonstrate heightened activation, sensitivity, and specificity towards their homologous antigens, making them a promising approach to enhance CAR therapy’s effectiveness against brain tumors [52].

In recent years, significant progress has been achieved in identifying tumor-specific antigens and advancing innovative CAR therapies for brain tumors. The exploration of targeting multiple antigens simultaneously holds great potential in addressing the challenges posed by tumor heterogeneity. Yet, further research is crucial to refine the selection and combination of targeted antigens for optimal treatment efficacy. Additionally, gaining a comprehensive understanding of the evolution of brain tumors post-CAR therapy, specifically analyzing changes at the single-cell level within intratumoral subsets, will provide valuable insights into overcoming tumor heterogeneity and enhancing treatment outcomes.

#### 3.1.2. Integrating Resistance Mechanisms into CAR T Cells

In the immunosuppressive microenvironment of tumors, cytokines like transforming growth factor β (TGF-β) hinder the antitumor function of CAR T cells. TGF-β, in particular, is secreted by both tumor cells and cells in the tumor microenvironment, promoting CAR T-cell depletion by upregulating PD1, resulting in the inhibition of their antitumor effects [56,57,58,59]. Researchers have adopted several strategies to overcome cytokine inhibition in the TME, particularly in the context of CAR T therapy. One approach involves the utilization of the CRISPR/Cas9 system to knock out cytokines or their receptor genes, thereby mitigating their immunosuppressive effects [58,59,60] (Figure 1b). Studies have demonstrated that CAR T cells with knockout of the TGFβRII gene exhibit improved tumor elimination in xenograft and transplanted solid tumor models. Specifically, TGFβRII KO CAR T cells decrease the conversion of Treg cells and prevent CAR T-cell depletion [58,60]. Moreover, TGFβRII KO CAR T cells experience reduced depletion upon repetitive antigen stimulation compared to wild-type CAR T cells. Additionally, in the presence of exogenous TGFβ, TGFβRII KO CAR T cells demonstrate enhanced proliferation, cytotoxicity, and cytokine secretion [60]. Another strategy involves the use of PD-1-knockout EGFRvIII CAR T cells, which significantly inhibit EGFRvIII-expressing GBM cells in vitro without altering the expression of T-cell phenotype or other checkpoint receptors [59].

CAR T cells can also counteract the suppressive action of cytokines by secreting corresponding antibodies (Figure 1c). For example, they can secrete α-PD-1 single-chain antibodies, effectively blocking PD-1 and enhancing T-cell activation in advanced tumors with strong immunosuppression. Compared to the wild type, CAR T cells producing single-chain antibodies exhibit reduced PD-1 expression, but increased expansion and cytotoxicity to solid tumors in subcutaneous and in situ xenograft models [61]. Additionally, CAR T cells can secrete PD-L1 single-chain antibodies that block PD-1/PD-L1 signaling in vitro and in NCG mouse xenograft cancer models, thereby enhancing the antitumor activity in solid tumors. Furthermore, in vivo autocrine PD-L1 single-chain antibodies greatly reduce CAR T-cell depletion [62].

#### 3.1.3. Incorporating Cytokine Support into CAR T Cells

In the context of solid tumor treatment, the remodeling of the tumor microenvironment (TME) is of paramount importance. Various cytokines have been identified as inhibitors of the antitumor function of CAR T cells. Cytokines, which are broadly categorized into regulatory, pro-inflammatory, and anti-inflammatory cytokines. Among these cytokines, IL7 in the regulatory cytokines and IL12 and IL15 in the pro-inflammatory cytokines have shown promise when utilized in fourth-generation CAR T cells [63,64,65]. The cells not only secrete pro-inflammatory cytokines to enhance their direct tumor-killing activity but also promote proliferation of endogenous T cells, while protecting themselves from immunosuppressive cytokines [17] (Figure 1d).

IL7 is a promising candidate for enhancing the antitumor function of CAR T cells. Its signaling in T cells promotes their survival, proliferation, and the formation of memory T cells [66,67,68,69]. IL7 has shown to be less toxic at certain doses compared to other cytokines like IL2, IL12, and IL15 in clinical trials [70,71,72,73]. To deliver IL7 locally, CAR T cells can be designed to secrete it. Studies have demonstrated that co-expression of IL7 and IL7 Flt3L leads to improved CAR T-cell quantity and overall survival in EGFRvIII heterogeneous tumors that were pretreated with non-lymphatic depletion irradiation compared to conventional or Flt3L-CAR T-cell therapy [63]. To further enhance the antitumor response of CAR T cells, IL7 can be designed to co-secrete with CCL. This approach has shown better therapeutic outcomes than traditional CAR T-cell therapy [74,75]. Co-expression of IL7 and CCL19 in anti-mesothelin CAR T cells downregulates the expression of T-cell depletion markers, such as PD-1 and TIGIT, thereby preventing lymphatic depletion [75]. Additionally, CAR T cells co-expressing IL12 and IFNα2 have demonstrated improved antiglioma activity in three in situ immunocompetent mouse glioma models, without any signs of toxicity, compared to T cells expressing CAR or cytokines alone. Co-expression not only promotes the pro-inflammatory tumor microenvironment but also reduces T-cell depletion [64].

Another crucial cytokine is IL15, which plays a pleiotropic role in improving the immune response to tumor cells and in the development, homeostasis, activation, and survival of T cells, natural killer cells (NK), and NK-T cells [76]. IL15-secreting IL13α2-CAR T cells effectively deplete IL15Rα-high MDSCs, which are essential components of the GBM microenvironment, and reduce the levels of their immunosuppressive molecules in vitro. A more effective approach involves fusing IL15 directly to the CAR in IL13Rα2-CAR T cells (CAR.IL15f). CAR.IL15f T cells exhibit a higher degree of reversal of the immunosuppressive TME, leading to improved survival in glioma models [65]. Moreover, GD2CAR IL-15-T cells were found to effectively penetrate the brain and control tumor growth after intravenous administration in an aggressive in situ xenograft model of glioblastoma [77].

### 3.2. Tumor Heterogeneity and Possible Solutions

#### 3.2.1. Tumor Heterogeneity

It is important to note that glioblastoma (GBM) exhibits high intratumoral heterogeneity, which can be observed through the sequencing of multiple-site samples of a tumor as well as single-cell sequencing [78,79,80,81]. This intratumoral heterogeneity not only contributes to resistance to chemotherapy or targeted therapy but also leads to short-term relapses, ultimately resulting in treatment failure even after CAR T-cell therapy. In addition, some targeted antigens may be lost during later stages of tumorigenesis following CAR T-cell infusion. For example, early clinical trials of IL-13Ra, EGFRvIII, and GD2-targeted CAR T-cell therapies have shown that this therapy can exert selective pressure on the specific antitumor activity of a single antigen, leading to antigen loss and ultimately tumor recurrence [23,25,28]. Therefore, overcoming intratumoral heterogeneity remains the key to improving efficacy in CAR design.

#### 3.2.2. Possible Solution: New Target Discovery

Brain tumors often exhibit tumor antigen heterogeneity, which can lead to drug resistance in response to CAR T-cell therapy. This highlights the need to expand the library of targeted antigens and identify optimal targets to prevent antigen escape in brain tumors. Aside from the targets discussed in previous sections, researchers are investigating other antigens for CAR T-cell therapy of brain tumors. For instance, B7H3 is an immune checkpoint molecule that favors tumor immune escape and is crucial in GBM cell differentiation, migration, and invasion [78,79,80]. Among all subtypes of tested GBM specimens, a remarkable 76% showed highly expressed B7H3 levels, consistently expressed in GBM-NS containing CSCs, while it was undetectable in normal brains [81]. Due to its high expression in tumor cells, the molecule B7-H3 has emerged as a highly attractive target for CAR therapy in GBM research. A completed preclinical trial demonstrated that B7-H3-directed CAR T cells effectively control tumor growth in vitro and xenograft models and are expressed upon tumor recurrence. Moreover, there have been no reported toxicities of B7-H3 regarding nonmalignant recognition [81]. Currently, the evaluation of B7-H3 CAR T cells in GBM patients is underway. Several clinical trials have already been conducted, where patients received a single infusion of B7-H3 CAR T cells or a combination therapy involving temozolomide. However, the clinical results from these trials have not yet been reported (NCT04385173, NCT04077866, and NCT05241392).

Integrin is a heterodimer adhesion receptor that is crucial in cell migration and tissue invasion [82]. Integrin αvβ3 is typically expressed in newly formed endothelial cells, including the tumor vascular system [83]. This expression is associated with many tumors, especially central nervous system (CNS) tumors, while being poorly expressed in normal tissues [84]. In a recent preclinical trial, it was observed that patient-derived GBM cell lines exhibited high expression of integrin αvβ3 on their surface. To target this integrin specifically, αvβ3 CAR T cells were designed and tested for their antigen-specific tumor cell-killing ability in both in vivo implanted DIPG and GBM tumor models as well as cytotoxicity assays. These CAR T cells showed potent tumor regression in situ, resulting in complete tumor elimination without any subsequent recurrence in GBM tumors, with detectable TCF-1+ αvβ3 CAR T cells even after tumor clearance. Notably, in this study, αvβ3 CAR T cells did not produce “on target, off tumor” kills in subjects. But this does not rule out the possibility of such toxicity in humans [84].

P32, also known as subcomponent-binding protein (C1QBP), has received significant attention as a target for CAR therapy. Its primary location is within the mitochondrial matrix, where it is involved in maintaining oxidative phosphorylation and regulating the synthesis of mitochondrial DNA-encoded genes [85,86,87]. Notably, P32 exhibits high levels of expression on the surfaces of tumor cells and tumor-associated endothelial cells, while being absent in normal brain tissue and other organs [88,89]. The expression of P32 correlates with the progression of the disease. Recent studies have indicated that hyaluronic acid (HA) can disrupt the normal localization of P32 in mitochondria, causing it to move to the cell surface. On the surface of tumor cells, p32 interacts with HA, which potentially contributes to the increased aggressiveness and proliferative capacity of GBM. Furthermore, P32 binds to integrin αvβ3 on the cell surface, leading to an upregulation of MT1-MMP expression at the transcriptional level. This activation further enhances tumorigenicity and cell migration through MMP-2. In preclinical trials, these specific CAR T cells targeting P32 displayed the ability to effectively identify and selectively eliminate glioma cells expressing P32, as well as tumor-derived endothelial cells in vitro. These cells also successfully suppressed tumor growth in situ and xenograft mouse models. Importantly, no signs of toxicity were observed following topical or systemic administration of P32-specific CAR T cells [89].

In addition, tumor-specific peptides have emerged as a promising avenue for CAR therapy in the diagnosis and treatment of cancer. Among them is chlorotoxin (CLTX), a 36-amino acid tumor-binding peptide derived from the venom of the *Leiurus quinquestriatus* scorpion [90]. CLTX demonstrates strong binding affinity for GBM and other neuroectodermal tumors, while exhibiting minimal reactivity with nonmalignant cells in the brain and other tissues [91]. Moreover, CLTX itself does not possess any cytotoxic effects on tumor or normal tissues [92,93]. A completed preclinical trial has confirmed that the incorporation of CLTX into a CAR construct redirects T cells to selectively identify tumors, resulting in minimal off-target effects on healthy cells or tissues [94]. In another preclinical trial, CLTX-EQ-28ζ CAR T cells were engineered with a CLTX tumor-targeting domain. These CLTX-CAR T cells exhibited potent anti-GBM activity, leading to tumor regression in an in situ xenograft GBM tumor model. Moreover, in patient-derived xenograft models, CLTX-CAR T cells effectively targeted tumors even with low expression levels of other GBM-associated antigens, such as IL13Rα2, HER2, or EGFR. This approach expands the targeted repertoire of CAR T cells and overcomes the challenge of tumor heterogeneity by providing comprehensive coverage of tumor cells. However, the potential immunogenicity of CLTX may impose limitations on its future clinical application [95].

The exploration of GBM and other brain tumor targets presents new possibilities for the advancement of CAR T-cell therapies. Building upon this foundation, it becomes imperative to thoroughly assess the safety of these neoantigens through clinical trials. Such evaluations are crucial in guaranteeing the effectiveness and safety of the therapies that target one or multiple antigens, thereby expanding their potential application across a diverse range of brain tumors and a broader spectrum of intratumoral cells.

### 3.3. Tumor-Suppressive Microenvironment and Possible Solutions

#### 3.3.1. Tumor-Suppressive Microenvironment

The immunosuppressive tumor microenvironment (TME) of glioblastoma consists of various components that contribute to the inhibition of immune responses against the tumor. One aspect of the immunosuppressive TME in glioblastoma is the presence of physical and metabolic barriers. These barriers can hinder the infiltration of immune cells into the tumor, limiting their access to tumor cells [96]. Additionally, the TME contains immunosuppressive cells, such as regulatory T cells, macrophages, and myeloid-derived suppressor cells. These cells create an immunosuppressive environment by releasing factors that hinder the activity of effector T cells and facilitate tumor growth [97]. Moreover, tumor-derived soluble factors contribute to glioblastoma’s immunosuppression. These factors include cytokines and chemokines that suppress pro-inflammatory responses and impede the proliferation and function of effector T cells. Furthermore, the absence of T-helper 1-associated cytokines, which are crucial for stimulating cell-mediated immune responses, further amplifies the immunosuppressive characteristics of the TME [98]. Interestingly, studies have shown that following CAR T-cell therapy targeting EGFRvIII in glioblastoma, there is an observed increase in the infiltration of regulatory T cells and production of immunosuppressive cytokines within tumor tissues [25]. This suggests that tumor cells activate adaptive resistance mechanisms to counteract the antitumor immune response induced by CAR T-cell therapy. Comprehending these immunosuppressive mechanisms within the glioblastoma TME is vital for devising approaches to overcome them and enhance the effectiveness of CAR T-cell therapies for this challenging type of cancer.

#### 3.3.2. Possible Solution: Combination with Oncolytic Viruses

Combination therapies that combine oncolytic viruses with CAR T cells have demonstrated significant potential in overcoming the suppressive tumor microenvironment and enhancing the persistence and function of CAR T cells. In the context of adjuvant therapy, oncolytic viruses have demonstrated promising combinations with CAR T-cell therapy in the TME of solid tumors [99,100]. The combination therapy primarily promotes tumor regression by creating a more pro-inflammatory environment and reducing anti-inflammatory factors. One preclinical study involved combining oncolytic herpes simplex virus 1 (oHSV-1) with CD70-specific CAR T cells. In this therapy, oHSV-1 infection of tumor cells transforms them into cytokine factories that stimulate the secretion of IFN-γ. This immune transformation changes the TME from immunosuppressive to immunostimulating, allowing CAR T cells to infiltrate and activate. In animal models of glioblastoma, this combination therapy has shown significant antitumor efficacy, leading to extensive infiltration of T cells and natural killer cells, as well as reduced levels of regulatory T cells and TGF-β1 [101]. While using oAD-IL7 alone has demonstrated moderate cytotoxicity in in vitro studies, it has not been able to induce a thorough and long-lasting antitumor therapeutic effect in vivo. However, when bound to CAR T cells, oHSV-1 enhances T-cell proliferation, reduces T-cell apoptosis, and results in prolonged survival in xenograft mice carrying tumors [102]. IL-21 and TTV have also been identified as highly effective immunomodulatory molecules and oncolytic viruses for suppressing solid tumors in mouse models, respectively. Therefore, a novel recombinant oncolytic virus called rTTVΔTK-mIL21 has been investigated and has shown significant regression of distant and proximal tumors following direct injection in mice. It selectively enriches immune effector cells, thus inducing systemic responses. Similarly to previous experiments, it exhibits significant synergistic effects when combined with CAR T cells in the treatment of brain tumors [103].

Oncolytic virus infection infects tumor cells and induces the expression of new epitopes on the cell surface. A nonsignaling truncated CD19 (CD19t) protein is expressed by selectively delivering the oncolytic virus encoding CD19t (OV19t) to tumors, allowing CD19-CAR T cells to target tumor cells. The infection of tumor cells leads to the presentation of new CD19 on the cell surface prior to virus-mediated tumor lysis. In several mouse tumor models, the combined administration of oncolytic viruses and CD19-CAR T cells promotes tumor control, inducing a local immune response characterized by the infiltration of T cells from both endogenous and adoptively transferred sources. The killing of tumor cells by CAR T cells also leads to the release of the virus from the dying tumor cells, facilitating the spread of CD19t expression within the tumor [104]. The effectiveness of combining oncolytic viruses and CAR T cells relies on intricate interactions among various factors, including the choice of oncolytic virus vectors, targets for CAR T cells, types of cancer cells, and the host immune response [105]. Thus, selecting the appropriate oncolytic virus, timing the infection, and targeting combination CAR T cells are critical for improving the efficacy of combination therapy.

#### 3.3.3. Possible Solution: Combination with Immune Checkpoint Inhibitors

Combining immune checkpoint inhibitors (ICIs) with CAR T-cell therapy shows promise as a way to overcome the inhibitory tumor microenvironment and enhance CAR T-cell function. Immune checkpoint molecules such as PD-1, CTLA-4, and PD-L1 have been extensively studied in this regard [106]. Preclinical studies have shown that blocking PD-1 or CTLA-4 in mouse models of glioblastoma increased long-term tumor-free survival and altered the inhibitory tumor microenvironment [107]. Clinical trials have further demonstrated that nivolumab, a PD-1 inhibitor, can enhance chemokine transcript expression and increase immune cell infiltration to support the local immunomodulatory effects of the treatment. However, the efficacy of ICI therapy is limited in some tumors due to a lack of preexisting antitumor immune responses [108]. Therefore, combining ICIs with immunotherapies that provide T-cell infiltration, like CAR T-cell therapy, may improve response rates. This combination therapy can alleviate lymphocyte depletion, prolong T-cell proliferation, sustain tumor-killing activity, and further improve the survival of patients with glioblastoma treated with CAR T cells. In particular, the co-delivery of CAR T cells and PD-1 checkpoint inhibitors has shown significant antitumor responses in glioblastoma [109,110]. In mouse models, the combination therapy enhanced the efficacy of humanized EGFRvIII CAR T cells by blocking PD-1 compared to using humanized EGFRvIII CAR T cells alone [110]. Additionally, PD-1 blockade of CAR T cells exhibited higher killing efficiency in vitro and increased the number of tumor-infiltrating lymphocytes (TIL) in mouse models [109]. Further research on the combination therapy of ICIs and CAR T cells is ongoing. In one clinical trial (NCT03726515), the effectiveness of combining CAR T cells with pembrolizumab (PD-1 inhibitor) is being evaluated in GBM patients. Furthermore, another trial (NCT404003649) is assessing the combination of IL13Rα2-CAR T cells with nivolumab (PD-1 inhibitor) and ipilimumab (CTLA-4 inhibitor).

## 4. Conclusions

CAR T-cell immunotherapy stands as a transformative approach in cancer treatment, offering promising potential for GBM therapy. While immunotherapy for GBM is still in its early stages, recent progress in neuroimmunology and cancer immunotherapy has sparked optimism about its therapeutic possibilities. Insights acquired from preclinical and clinical studies on CAR T cells serve as the basis for future trial designs. Early trial outcomes highlight such challenges as the suppressive tumor microenvironment and tumor heterogeneity, emphasizing the need to address these hurdles for improved CAR T-cell therapy efficacy. Targeted experiments have yielded promising therapeutic results, offering potential solutions to overcome treatment obstacles.

## Figures and Tables

**Figure 1 cancers-15-05652-f001:**
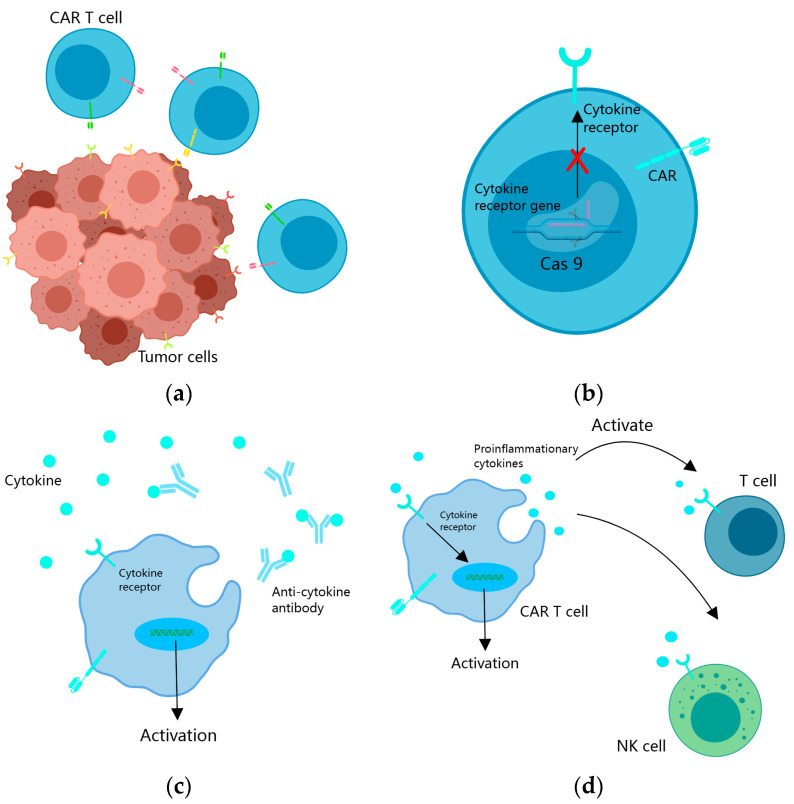
Strategies to overcome tumor heterogeneity and an immunosuppressive microenvironment. (**a**) CAR T cells are engineered to target multiple tumor antigens simultaneously to overcome tumor heterogeneity; (**b**) cytokine receptor genes in the nucleus of CAR T cells are knocked out through gene editing to resist an immunosuppressive microenvironment; (**c**) CAR T cells produce specific antibodies to counteract the effects of cytokine inhibition; (**d**) CAR T cells enhance the direct antitumor effect of CAR T cells and intrinsic antitumor effect by secreting pro-inflammatory cytokines and regulatory cytokines.

**Table 1 cancers-15-05652-t001:** Complete clinical trials of CAR T cells for GBMs.

Target Antigen	CAR T Product	Phase	Study Population	Dosage of CAR and Route of Delivery	Patient Response	NCT Identifier
IL13Rα2	IL13(E13Y)-zetakine+ CD8+ CTL clones	Phase I	Recurrent glioblastoma (n = 3)	11–12 intracavitary infusions of 10^7^–10^8^ CAR T cells or 5 intratumoral infusions of up to 10^8^ CAR T cells.	Median OS: 10.3 months	NCT00730613 [23]
	IL13(E13Y)-41BBζ memory-derived T cells	Phase I	Recurrent glioblastoma (n = 1)	6 intracavitary injections or 10 intraventricular injections of up to 10 × 10^6^ IL13Rα2-directed CAR T cells	CR: 7.5 months	NCT02208362 [24]
EGFRvIII	EGFRvIII-41BBζ T cells	Phase I	Recurrent glioblastoma (n = 10)	Single intravenous injection of 1.75–5 × 10^8^ CAR T cells	Median OS: 8.4 months	NCT02209376 [25]
	EGFRvIII-CD28-41BBζ T cells	Phase I/II	Recurrent glioblastoma (n = 18)	Single intravenous infusion of 6.3 × 10^6^–2.6×10^10^ CAR T cells	Median OS: 6.9 months, median PFS: 1.3 months,	NCT01454596 [26]
HER2	HER2-CD28ζ VST	Phase I	Progressive glioblastoma (n = 17)	1–6 intravenous infusions of 10^6^–10^8^/m^2^ CAR T cell	Median OS: 11.1 months	NCT01109095 [27]
GD2	GD2-CD28-41BBζ T cells	Phase I	Recurrent or progressive glioblastoma (n = 8)	Single intravenous infusion of 3 × 10^7^–2.1 × 10^8^ CAR T cells or one intracavitary infusion of 2.6–6.4 × 10^6^ CAR T cells	Median OS: 10 months	NCT03170141 [28]
EphA2	EphA2-41BBζ T cells	Phase I	Recurrent glioblastoma (n = 3)	Single intravenous infusion of 1 × 10^6^/kg CAR T cells	Median OS: 5.5 months	NCT03423992 [22]

Abbreviation: CAR: chimeric antigen receptor, CR: complete remission, OS: overall survival, PFS: progression-free survival.

**Table 2 cancers-15-05652-t002:** Ongoing clinical trials of CAR T cells involving GBMs.

Target	Phase	Study Status	Condition	Interventions	NCT Identifier
IL13Rα2	Phase I	Active, not recruiting	Recurrent glioblastoma	IL13Rα2-41BB TN/MEM cells or IL13Rα2-41BB T lymphocytes	NCT02208362
	Phase I	Recruiting	Recurrent glioblastoma and glioma	IL13Rα2-41BB T lymphocytes	NCT04661384
	Phase I	Recruiting	Relapsed/refractory glioblastoma	IL13Ralpha2-41BB TN/MEM cells combined with or without ipilimumab and nivolumab	NCT04003649
EGFRvIII	Phase I	Unknown	Glioblastoma	EGFRvIII CAR T cells, cyclophosphamide and fludarabine	NCT02844062
	Phase I	Unknown	Glioblastoma and malignant glioma	EGFRvIII-41BB CAR T lymphocytes	NCT05063682
	Phase I	Not yet recruiting	Recurrent glioblastoma	EGFRvIII CAR T cells	NCT05802693
EGFR and IL13Rα2	Phase I	Recruiting	Glioblastoma	EGFR-IL13Rα2 CAR T Cells	NCT05168423
B7-H3	Phase I	Recruiting	Relapsed/refractory glioblastoma	B7-H3 CAR T cells combined with temozolomide	NCT04385173
	Phase I/II	Recruiting	Relapsed/refractory glioblastoma	B7-H3 CAR T cells combined with temozolomide	NCT04077866
	Phase I	Recruiting	Relapsed/refractory glioblastoma	B7-H3 CAR T cells	NCT05366179
	Phase I	Recruiting	Relapsed/refractory non-brainstem primary CNS tumors and brainstem high-grade neoplasms	B7-H3 CAR T cells	NCT05835687
	Phase I	Recruiting	Recurrent glioblastoma	B7-H3 CAR T cells	NCT05241392
	Phase I	Recruiting	Recurrent glioblastoma	B7-H3 CAR T cells	NCT05474378
NKG2D	Phase I	Recruiting	Hepatocellular carcinoma, glioblastoma, medulloblastoma and colon cancer	NKG2D CAR T cells	NCT05131763
	Phase I	Withdrawn	Hepatocellular carcinoma, glioblastoma, medulloblastoma and colon cancer	NKG2D CAR T cells	NCT04270461
MMP2	Phase I	Recruiting	Recurrent or progressive glioblastoma	Chlorotoxin-CD28 CAR T-lymphocytes and chlorotoxin-CD28 CAR T lymphocytes	NCT04214392
	Phase I	Recruiting	Recurrent glioblastoma and glioma	CHM-1101 CAR T cells	NCT05627323
CD147	Phase I	Unknown	Recurrent glioblastoma	CD147 CAR T cells	NCT04045847
CD44 and CD133	Phase I	Not yet recruiting	Recurrent glioblastoma	IL7Rα modified CAR T lymphocytes.	NCT05577091

## Data Availability

The data presented in this study are available in this article.

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
