# Peer review of "Chimeric Antigen Receptor T-Cell Therapy for Glioblastoma"

_cancers, 2023, doi:10.3390/cancers15235652_

Round 1

Reviewer 1 Report

Comments and Suggestions for Authors

The review article by Kun Ma and Ping Hu on CAR-T cell therapy for GBM is timely with the shifting focus of CAR-T cell therapy towards solid organ tumours. There are certainly more challenges using CAR-T cell therapy in these tumours and particularly with GBM as highlighted by this article.  The article was overall well written and covered a breadth of current approaches with CAR-T cell therapy in GBM. 

The authors discussed several clinical trials using different target antigens (Table 1 and 2). Interestingly, these trials have reported different modes of CAR-T cell delivery ie intravenous, intracavitory, or intraventricular. It would therefore be relevant to discuss the rationale, benefit and challenges for the different approaches.

Line 272 – The authors discuss the role of PD-1 KO genes in an EGFRvIII CAR-T construct to block the inhibitory PD-1/PD-L1 axis in the GBM tumour microenvironment. This mirrors the concept of using immune checkpoint inhibition to augment CAR-T responses which has been previously explored with haematological malignancies. Notably, the use of immune checkpoint inhibitors with CAR-T cell therapy is also currently being investigated in a GBM clinicsl trial eg: NCT03726515 which is investigating EGFRvIII CAR-T with Pembrolizumab. This would be also relevant to include in the manuscript discussion.

Line 350 – The authors mention clinical trials investigating a novel antigen ‘B7-H6’ but the trials listed report targeting the B7-H3 antigen. Please review for a typographical error.  

Author Response

Comments 1: [The authors discussed several clinical trials using different target antigens (Table 1 and 2). Interestingly, these trials have reported different modes of CAR-T cell delivery ie intravenous, intracavitory, or intraventricular. It would therefore be relevant to discuss the rationale, benefit and challenges for the different approaches.]

Response 1: Thank you for your feedback. We appreciate your agreement and would like to address the advantages and disadvantages of systematic and local delivery methods for CAR T cells in glioma based on previous clinical trials. Considering the unique characteristics of gliomas and the presence of the blood-brain barrier, local delivery of CAR T cells has shown to be more effective compared to systemic delivery methods such as intravenous infusion. This localized approach allows for direct targeting of tumor cells within the brain, maximizing the therapeutic effect of CAR T cells. This change is made in page 7, line 215.

“[updated text in the manuscript if necessary]”

Comments 2: Line 272 – The authors discuss the role of PD-1 KO genes in an EGFRvIII CAR-T construct to block the inhibitory PD-1/PD-L1 axis in the GBM tumour microenvironment. This mirrors the concept of using immune checkpoint inhibition to augment CAR-T responses which has been previously explored with haematological malignancies. Notably, the use of immune checkpoint inhibitors with CAR-T cell therapy is also currently being investigated in a GBM clinicsl trial eg: NCT03726515 which is investigating EGFRvIII CAR-T with Pembrolizumab. This would be also relevant to include in the manuscript discussion

Response 2: Agree. We agree with your perspective. Combining immune checkpoint inhibitors with CAR T cells is promising in overcoming the inhibitory tumor microenvironment and enhancing CAR T cell function. We briefly outline the current status of CAR T cell combination therapies with immune checkpoint inhibitors and provide a summary of relevant clinical trials in our discussion. This change is made in page 14, line 546.

“[updated text in the manuscript if necessary]”

Comments 3: Line 350 – The authors mention clinical trials investigating a novel antigen ‘B7-H6’ but the trials listed report targeting the B7-H3 antigen. Please review for a typographical error.

Response 3: We sincerely apologize for the error in the initial manuscript and appreciate you bringing it to our attention. The bug has been rectified and we are committed to ensuring thorough inspections in future manuscripts. This change is made in page 11, line 418.

“[updated text in the manuscript if necessary]”

4. Response to Comments on the Quality of English Language

Point 1:

Response 1:   Thank you for your comments about the quality of our English. We have made some changes to make the English quality better. 

Reviewer 2 Report

Comments and Suggestions for Authors

The authors present a review of existing research in CAR T therapy. The review is a very helpful read for the researcher entering the field. 

It will be helpful if the authors include an illustration of the existing therapeutic targets discussed in the paper to understand the current state. The authors also suggest future strategies and the challenges to the existing CAR T therapy. It will be very helpful to include an illustration for the suggested approach (inclusion of cytokines) and how this approach may be helpful in improving patient outcomes.

Author Response

Comments 1: It will be helpful if the authors include an illustration of the existing therapeutic targets discussed in the paper to understand the current state.

Response 1: Thank you for your response. The existing tables are effective in providing a comprehensive overview of the current status of therapeutic targets for GBM, encompassing both completed and ongoing clinical trials in recent years. We have also added some preclinical trials in GBM. And there are existing preclinical trials in other parts.  Tables may be appropriate to show the state of existing therapeutic targets. This change is made in page 2, line 76.

“[updated text in the manuscript if necessary]”

Comments 2:. The authors also suggest future strategies and the challenges to the existing CAR T therapy. It will be very helpful to include an illustration for the suggested approach (inclusion of cytokines) and how this approach may be helpful in improving patient outcomes

Response 2: Thank you for your agreement. We completely understand the importance of using illustrations to visually explain complex mechanisms. Therefore, we design a figure which illustrate the action of CAR T cell design in overcoming tumor heterogeneity and suppressive tumor immune microenvironment. This will be helpful for readers' understanding and interpretation of the content. This change is made in page 8, line 427.

“[updated text in the manuscript if necessary]”

4. Response to Comments on the Quality of English Language

Point 1:

Response 1:   Thank you for your comments about the quality of our English. We have made some changes to make the English quality better. 

Reviewer 3 Report

Comments and Suggestions for Authors

Comments:

Major Comment

1. While author described CARs against human GBM, there are a lot of preclinical GMB studies associated with mouse CARs which was highly expected in this review.

2. While an enhancement of CAR construct can improve the antitumor activity of the CAR, it may not be sufficient to overcome the limitations of the immunosuppressive TME. Therefore, role of Combinatorial Therapy to Enhance Car T Cell Efficacy could be another major strategy to treat GMB with CARs.

3.     Author should describe the limitation of CAR therapy against GMB.

Minor Comment:

1.     Section 2.2 While authors compiled and ongoing clinical trials associated with CARs treatment in glioblastoma, however they did not mention some of the important ongoing clinical studies associated with CARs in glioblastoma, these include B7-H3, CD147, MMP2, NKD2D ligand.

2.     Section 2.2.1: Author showed IL13Ra2 is highly expressed specifically on glioblastoma but not in the normal brain tissues, however they didn’t the mention about IL13 and/or IL13Ra1. It would have been better if they describe the why IL13Ra2 was used in first place.

3.     2.2.2 Please mention and cite reference of preclinical model of EGFRvIII specific CAR.

4.     2.2.3 Author mentioned positive outcome associated with HER2 targeted CARs. However, HER2 is also expressed in normal tissue. Author should cite the reference describing the clinical study associated where HER2 was targeted with CARs and high accumulation of CARs were found in normal tissue where HER2 is as also expressed.

Comments on the Quality of English Language

additional scientific editing will help

Author Response

Comments 1: While author described CARs against human GBM, there are a lot of preclinical GMB studies associated with mouse CARs which was highly expected in this review

Response 1: Thank you for your feedback. We agree with you and we've added a section on preclinical trials of CAR T cells in mice. We focus on the formation and role of mouse models and the role of CAR T cells in mouse. This change is made in page 2, line 76.

Comments 2: While an enhancement of CAR construct can improve the antitumor activity of the CAR, it may not be sufficient to overcome the limitations of the immunosuppressive TME. Therefore, role of Combinatorial Therapy to Enhance Car T Cell Efficacy could be another major strategy to treat GMB with CARs.

Response 2: Agree with you, CAR T cell therapy can better eliminate tumor cells and overcome the barriers of tumor heterogeneity and suppressive immune microenvironment by combining with other immunotherapies, such as immune checkpoint inhibitors, to form a synergistic effect. This change is made in page 14, line 546.

Comments 3: Author should describe the limitation of CAR therapy against GMB.

Response 3: We agree with you. We describe the limitations of the above clinical trials, including tumor heterogeneity, inhibitory immune microenvironment, on-target off-tumor effect, etc. Combining it with two sections (the tumor suppressive microenvironment and tumor heterogeneity) can provide a better understanding of the limitations of CAR T cells.  his change is made in page 7, line 230.

Comments 4: Section 2.2 While authors compiled and ongoing clinical trials associated with CARs treatment in glioblastoma, however they did not mention some of the important ongoing clinical studies associated with CARs in glioblastoma, these include B7-H3, CD147, MMP2, NKD2D ligand..

Response 4: Thank you for your feedback. We summarize the ongoing clinical studies of these targets in a table. At the same time, the pre-clinical studies of B7-H3 are described in the following sections (line 409). Since several other targets have not had new results in recent years, we do not focus on describing their current status.

Comments 5: Section 2.2.1: Author showed IL13Ra2 is highly expressed specifically on glioblastoma but not in the normal brain tissues, however they didn’t the mention about IL13 and/or IL13Ra1. It would have been better if they describe the why IL13Ra2 was used in first place.

Response 5: We agree with you that IL13 can bind to both receptors at the same time, and since it binds more easily to high-affinity receptors, IL13Rα2. It is an appealing target for CAR T cell therapy. This change is made in page 5, line 121.

Comments 6: 2.2.2 Please mention and cite reference of preclinical model of EGFRvIII specific CAR.

Response 6: We appreciate your feedback. We have highlighted several preclinical studies about EGFRvIII specific CAR T cells behind its clinical trials. This change is made in page 6, line 163.

Comments 7: Author mentioned positive outcome associated with HER2 targeted CARs. However, HER2 is also expressed in normal tissue. Author should cite the reference describing the clinical study associated where HER2 was targeted with CARs and high accumulation of CARs were found in normal tissue where HER2 is as also expressed.

Response 7: We agree with you that the target is expressed on both normal tissues and tumor cells. Therefore, we focus on describing the serious adverse effect in clinical trials. This change is made in page 6, line 175.

4. Response to Comments on the Quality of English Language

Point 1:

Response 1:   Thank you for your comments about the quality of our English. We have made some changes to make the English quality better.